# Energy Performance Certificates and Its Capitalization in Housing Values in Sweden

**Mats Wilhelmsson** 

Building and Real Estate Economics, Royal Institute of Technology, 114 28 Stockholm, Sweden;
mats.wilhelmsson@abe.kth.se; +46-(0)-768-675-280

**Abstract:** The impact on energy performance certificates on housing prices has been investigated extensively in recent years. However, the results of these investigations are mixed. We add to the literature by more specifically controlling for potential biases, by employing a combination of alternative approaches to estimate the causal relationship between house prices and energy performance certificates. We use a traditional hedonic modeling approach, but we additionally employ propensity score methods to be able to compare treated houses with a control group. We also investigate the impact of the outliers, spatial dependency, and parameter heterogeneity of our estimates. Moreover, we use the quantile regression technique to test the hypothesis that the capitalization effect varies across the price distribution. Our results, analyzing more than 100,000 observations, indicate there is an upward bias if one is not controlling for outlier and selection bias. Regardless of the propensity score method approach, the results are lower than a model (around 3 percent capitalization, compared to 6 percent). However, our results do not support that the impact of energy performance certificates varies in the price distribution. Consequently, the certificates are not differently capitalized in the high-end housing price segment. Finally, our results support the hypothesis that the energy performance certificate should be more capitalized into house prices in the northern and colder parts of Sweden than in the southern regions.

**Keywords:** energy performance certificates; climate; hedonic price equation; selection bias; propensity score method

---

## 1. Introduction

The importance of housing in general, and housing energy consumption, on climate change cannot be underestimated. According to some estimates, the building sector accounts for as much as 40 percent of total energy consumption [1]. Energy consumption can be derived from the construction of the building, but as much as 70 percent of the energy consumption comes from the operation of the buildings (see, e.g., [2]). According to statistics from European Statistical Office (Eurostat) [1], the residential sector represents 25 percent of final energy consumption in the EU.

Moreover, on average, 65 percent of final energy consumption in the residential sector goes to heating [1]. Energy cost is, therefore, a large proportion of household disposable income. In many countries, energy cost as a percentage of total household expenditures has increased. For example, the European Commission has estimated the share of income spent on energy to be around six percent in Denmark; the share is moderately higher in the Netherlands [3]. Statistics Sweden has estimated that household living cost as a percentage of disposable income is around 23 percent in 2014, and 5 percent of that is due to energy cost [3]. The relationship between household income level and energy expenditure as a share of income is negative. Consequently, long-term energy efficiency policies like energy performance certificates or energy labeling of houses can play an essential role in decreasing

greenhouse gas emissions from the residential sector but also potentially reduce the overall cost of energy for households.

Energy performance certificates (EPC) were introduced in the European Union in early 2000 to support reaching energy efficiency targets (the Energy Performance of Buildings Directive 2002/31/EC1; 2010/91/EU2). Sweden has had an EPC database since 2007 [4], and according to [5], 82 percent of the buildings in Sweden have an EPC. An EPC provides information on the energy consumption of the building. An independent certified energy expert produces it upon request by the owner of the building, and the EPC is valid for ten years. In Sweden, it is a governmental body (Boverket) that is in control of qualified experts' accreditation. An onsite visit is a mandatory requirement to issue an energy performance certificate [4]. The EPC comprises information about energy consumption for heating, cooling, and water, the electricity in the building, overall energy performance measured in kilowatt-hour (kWh) per square meter, as well as the heating and ventilation systems. To compare buildings with each other, an energy classification, such as EPC grade, is included in the certificates. The A–G rating scale is like those used for refrigerators and other electrical appliances. Energy grade A stands for low energy consumption (50 percent of the requirement for a new building), and G stands for high energy consumption (235 percent of the requirement for a new building).

The EPC undoubtedly has the potential to become a vital piece of valuable information for the consumer of housing and should, therefore, be capitalized into housing values. The author of [6] presented a comprehensive survey done on German homeowners about the possible impact of EPC but argues that it typically has only minor importance in the purchasing situation. They claim that one of the reasons for this is that only a small part of the dwellings in Germany, at the time, had an EPC grade. Moreover, they argue that homeowners do not trust the rating, and it is difficult for the potential buyer to grasp the financial implication of the EPC. To conclude, they argue that energy performance is a housing characteristic of minor importance. More important are, for example, location and size. Trusting the EPC grade is, of course, central, and consequently, the quality of the EPC is essential for trusting the grade. For example, the authors of [7] investigated the quality of the EPC itself. They concluded that there was room for improvement. They even argued that EPC did not add any novel information for the potential buyer and, therefore, the price impact should be limited.

Over recent years, research on the EPC capitalization effect on housing prices has been quite substantial. Most recent articles, for example, are [8] in Denmark, [9] in Wales, [10] in Sweden, and [11] in Spain. The results, however, have been mixed. Some find substantial price premiums for high grades, while others find no premium at all.

Hyland et al. [12] investigated the housing market in Ireland, and their results suggest that an A-rated house receives a price premium of 9.3 percent compared to a D rating. Moreover, a B rating increases the price by 5.2 percent. The authors of [8], for example, investigated the market response of the EPC grades on housing prices in Denmark over the period 2007–2012 with a data set of 117,000 observations, including house prices and housing characteristics. The price premium for the rating A/B is around 6.6 percent (compared to the default rating of D). Moreover, the authors of [9] investigated the impact of EPC grades using 192,000 observations in Wales over the period 2003 to 2014. The estimated price premium is highly statistically significant, with a price premium of as high as 12–13 percent for the rating A/B (compared to the default rating of D).

On the other hand, the author of [10] investigated the impact of energy performance on the Swedish single-family house prices in 2009 and 2010. Her result indicated there is no price premium at all. The author of [13] also investigated the Swedish housing market. Her results showed that energy performance has a positive impact on house prices. However, she only used a limited sample of 1000 observations. The results of [14] (also concerning Sweden) suggested that energy performance mainly has an impact on properties built before 1960 and in the low-price housing segment. The authors of [11] investigated the price of energy efficiency in the Spanish housing market using a data set of 1500 transacted homes. The explanation power of their hedonic model is on the lower side, and the estimated parameter concerning A–D EPC label is equal to a price premium of 5.4 percent. A recent

article by [15] analyzed the Finish apartment market. Their findings suggest a premium of 3.3 percent for apartments with an A/C energy performance rating. Even after controlling for maintenance costs, including energy costs, a premium of 1.3 percent is present, that is, the EPC rating is capitalized more than expected. The authors describe this as a green signaling effect.

The authors of [16] investigated the "Energy Star" certification on residential homes in Florida using data from 1997 to 2009. They showed there exists a price premium for new homes, but the effect is smaller after a couple of years as the property ages, contradicting the results of [14]. For newly built houses, the price premium for Energy Star is about 1.2 percent. Recently, the authors of [17] also analyzed the US housing market, looking at whether Energy Star certification provided any impact on sale prices in the Atlanta housing market between 2007 and 2010. The sample consisted of around 12,000 homes, of which 343 were certified as green homes. Their estimations showed that homes that earned an energy certificate were sold with a premium of almost 12 percent, which is substantially higher than estimated in [16].

As observed, the estimated impact of energy performance is positive in most cases, but the effect varies a lot. The purpose of the present paper was to control for potential biases that can affect some of the earlier articles. That said, in the present paper, the key objective was to estimate the willingness to pay for high-energy performance in housing. That is, our primary focus was the causal relationship between house prices and EPC. Estimating the willingness to pay for energy classification is essential in many respects. Firstly, it is important to clarify whether the classification is given the effect that is expected. If not, then the authorities may need to adapt and change how to present energy information in, for example, descriptions of the property on sale. Secondly, it is, of course, a piece of valuable information for both sellers and prospective buyers of a property. The estimation of marginal willingness to pay is an essential component in order to make decisions that make society more energy-smart. Our main contributions are: (1) evaluating data over a more extended period than earlier studies and in different climate zones, (2) controlling for the impact of outliers more effectively in the data set, (3) controlling for selection bias using the propensity score method, (4) controlling for spatial dependency, and (5) estimating the impact in the price distribution using quantile regression.

The structure of the paper is as follows: Section 2 presents the theoretical framework and used methodology, that is, the hedonic theory and the propensity score method. Section 3 presents the empirical analysis; finally, we summarize and conclude in Section 4.

## 2. Methodology

The conceptual framework used is the classical hedonic theory introduced by [18]. Essentially, a hedonic equation is a regression of house prices against housing attributes and neighborhood characteristics that determine these prices. We can interpret the regression coefficients as estimates of the implicit prices of these attributes, that is, the marginal willingness-to-pay. Numerous articles have been published using the hedonic price model. In addition to including housing attributes in our model, we included an indicator for the house's energy performance. It is not an attribute in the sense that it is good; instead, it is an indicator measuring the quality of the house concerning energy consumption. The estimated parameters concerning the energy performance can be interpreted as the willingness to pay for it.

The focus here is the unbiased estimation of the hedonic price equation; however, some problems can create biases in the estimated parameters. One problem in large data sets and with many used variables is that it is difficult to detect potential outliers. Another problem is endogeneity issues caused by selection bias or reversed causality. Another problem is the presence of spatial dependence and parameter heterogeneity. In addition, other problems that may occur are omitted-variable bias, heteroscedasticity issues, and multicollinearity.

As the authors of [19] emphasized, in analyzing data, applied researchers need to consider robust estimation methods for, for example, outliers. The authors of [17] also highlighted the importance of eliminating the influence of outliers. They removed outliers based on two standard deviations from

the mean of the dependent variable price and the independent variable size. However, the impact of outliers on estimated parameters is more complex than that. Therefore, as a pretest of the data used in the study, we detected outliers with several different measures. We followed the process laid out in [20]. In the first step, we estimated a hedonic price equation; firstly, we detected outliers with Cook's D and then analyzed the absolute residuals. That is, the most influential observations were excluded, and then observations with large absolute residuals were weighted down by an iterative process where observation weights are recalculated until convergence (see [21,22]). Berk [23] provides a full description of the methodology. In the present study, weights smaller than the 25th percentile of the distribution of the weights were excluded in the proceeding analysis.

Many of the preceding cited articles analyzing the impact of EPC on house prices adopted the hedonic framework. Just a few of them addressed the problem of selection bias. Selection bias means there is a risk that there is a different probability that the observations are included in the sample. Another related endogeneity problem comes from the fact that high-price areas might attract housing investment with high EPC grades. That is, the causality goes in both directions. The propensity score method addresses these problems.

Therefore, in the following step, we analyzed the potential impact of selection bias with the propensity score method. Rosenbaum et al. [24] proposed propensity score matching as a method to reduce the bias in the estimation of treatment effects with nonrandomized data sets. The main idea is that matching treated (EPC of A/C rating) with control observations (EPC D/G rating) reduces the bias by controlling the observed heterogeneity between the treated group and the control group. The propensity score method has been used in this context in, for example, [25,26]. Jensen et al. [8], however, addressed the selection bias problem differently. Denmark was one of the first countries that implemented EPC. By dividing the period before and after the public display of the rating in 2010, they were able to investigate the effect of it using a difference-in-difference approach. Their results indicated that the impact of the grade was relatively modest before 2010 and more substantial after 2010. Regrettably, the Swedish data we have access to makes it is impossible to adopt this approach.

In the analysis, we estimated the propensity scores with logistic regression (logit model) (pscore in Stata (see [27])). The effectiveness of the methods depends on how well we can control for the heterogeneity between the treatment and control group. The balancing property states that the propensity scores and characteristics between the groups should be equal. We, therefore, tested for this.

We used three different approaches to adjust the hedonic price equation (see, e.g., [28] for a useful review of different propensity score approaches). First, we used a multivariate approach where the propensity scores are used as either a covariate, the same way as the authors of [12] used Mill's ratio, or as in [29], where the inverse of the propensity score was used as sample weights. Reichardt [29] utilized this method when analyzing the effect of a rent premium on Energy Star- and LEED-certified buildings. Hirano et al. [30] demonstrated that weighting by the inverse of the propensity score leads to an efficient estimate of the treatment effect. Second, we used a propensity score matching, where we used nearest neighbor matching and radius matching. Only matched observations are used in the hedonic price equation (a similar approach in the real estate context has been used in [31–33]). Third, we used stratification matching, where a number of strata are constructed based on the distribution of the propensity scores. We used the strata as fixed effects in the hedonic price equation.

Quantile regression was used to test the hypothesis that the capitalization effect of the EPC varies in the price distribution. The estimation of quantile regression models aims to examine whether house prices, in the presence of high EPC grading, display an asymmetric behavior across the price distribution. The quantile regression model is, in this respect, more flexible than ordinary least squares (OLS). For a description of the method, see earlier work by [34]. For examples of recent applications of quantile regression models, see [35–42], as well as two recent papers by [43,44]. All these articles indicate that quantile regression analysis is suitable for a segmented housing market. The authors of [45] were one of the first to use quantile regression in a hedonic modeling framework. The authors concluded that some of the variations in estimated hedonic implicit prices derive from the fact that

housing attributes are not priced the same across the price distribution. Hence, it is of interest to analyze whether the EPC effect is equal across the price distribution. The quantile regressions were estimated using only matched samples and without influential observations (outliers).

Finally, we controlled for spatial dependency with spatial autoregressive models (see, e.g., [46], for an overview of the spatial hedonic models). The authors of [9] addressed the spatial dependency problem by including fixed effects (postal code) in the hedonic price equation. However, solving the spatial dependency problem by only adding fixed location effects may be insufficient to reduce the problem. We, therefore, estimated a spatial error model (SEM), spatial autoregressive model (SAR), and a spatial Durbin model (SDM) to find out whether we have any bias in our results. We implemented two variations of inverse distance as the spatial weight matrix. We followed the testing procedure of [47] by first estimating OLS; after that, we used Moran's I to test for spatial dependence. In the following step, we estimated the SDM and used the log-likelihood ratio test (LR-ratio) to test SDM against SEM and SAR. The spatial weight matrix is distance-based, using inverse distance and inverse distance squared. The choice of spatial weights matrix is based on goodness-of-fit measures like $R^2$, Akaike Information Criterion (AIC), and Bayesian information criterion (BIC).

Omitted variable bias was addressed in a couple of published papers. In the present study, one omitted variable represents the quality of the house, which is potentially highly correlated to EPC. Houses with a high EPC grade are most certainly also of higher quality in other aspects. Here, the omitted variable will cause an upward bias in our estimate of EPC grade. To some extent, the quality is correlated with age that is used in the propensity score matching. Hence, the problem should be of lesser magnitude. In the literature, omitted variable bias has been handled by restricting the sample in various ways. The author of [10] limited the data set into several submarkets (age groups and regions) to test the robustness of the estimates. The authors of. [9], on the other hand, restricted the data set by only including data from 2000.

The author of [10] included the number of days with frost per year as one of the covariates into the hedonic price equation. The impact of climate on house prices is highly significant. However, they did not test whether climate influences the capitalization of EPC into house prices. We tested the robustness of the estimates (parameter heterogeneity) through a spatial drift model that tests whether estimated parameters are constant in space. The author [17] addressed parameter heterogeneity with a spatial drift model across school districts. In a high-quality school district, the green housing certificates have a higher estimated impact on house prices (from 7.4 percent in low-quality school districts to 16.3 percent in high-quality school districts). We hypothesized that in colder areas in the northern part of Sweden (where the energy cost is higher), a house with a higher EPC grade should be capitalized into house prices to a higher degree.

## 3. Empirical Analysis

In the empirical section, we first present the data utilized in the study. After that, in step 1, we pretest the data for outliers before we present descriptive statistics in step 2. Propensity scores are estimated in step 3, together with two different matching approaches before the final hedonic price equation is estimated in step 4 using the program Stata. In step 5, the capitalization effect is estimated both on the average and across the price distribution. In step 6, we test the robustness of the estimates by estimating a spatial lag model and a spatial drift model. Further, in step 7, we test whether the capitalization is higher in the northern part of Sweden.

*Data*

We used data from Sweden covering the period between 2013 and 2018. In all, we were able to utilize more than 100,000 single-family housing arm's-length sales in all municipalities in Sweden. The data are from Mäklarstatistik AB, which is an association of brokers that covers about 80 percent of all brokered properties.

The dependent variable is the natural logarithm of house prices used, based on the transaction price when the contract was signed. The variables included in the data are, in addition to price (restricted to higher price than 340,000 SEK, equivalent to 35,000 USD), the independent variables EPC, living area (less than 300 square meters), number of rooms (less than 10 rooms), plot size (larger than 1000 square meters and smaller than 25,220 square meters), age, and contract date (sale year after 2012), as well as information about location measured by longitude and latitude.

Our key variable is EPC grade. To make our results somewhat comparable to some of the earlier studies, we tested the hypothesis of a price premium of EPC grade A–C to the default grade D–G. That is to say, A–C is the treatment, and D–G is the control group, if one employs the authors of [24] terminology. Hence, the variable EPC is a binary variable, where one is equal to the treatment group, and zero is equal to the control group.

*Step 1: Pretest of data*

The preliminary hedonics price equation is estimated, and the regression observation weights are calculated. As said, the natural logarithm of the price was used as a dependent variable, and the included co-variates are EPC (binary variable indicating one for A–C grade), the natural logarithm of the living area and age, number of rooms, plot size, and location, as well as the fixed monthly and municipality effects. The coefficients concerning the default hedonic model are presented in Table 3. Figure 1 shows the distribution of the estimated regression observation weights exhibited.

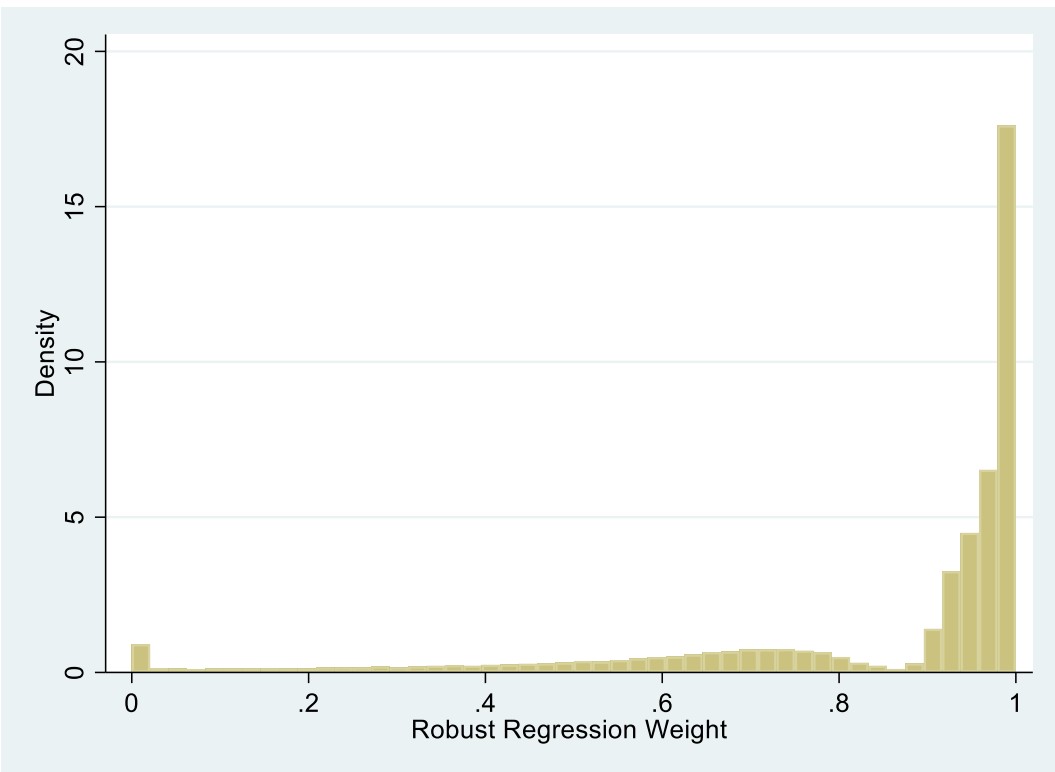

**Figure 1.** Estimated regression weights.

The average regression weight is equal to 0.9, and the median is equal to 0.96. The standard deviation of the regression weights is equal to 0.13. At the percentile 0.25, the regression weight is equal to 0.75, which we used as a cut-off value. After deleting potential outliers (around 20,000 outliers or 20 percent of the original data), the data set consisted of almost 83,000 transacted single-family houses after the deletion of the outliers. Around 19,000 of them were treated, that is, with an EPC grade of A–C, and around 63,000 were in the control group.

*Step 2: Descriptive analysis*

Figure 2 exhibits the distribution of EPC grades in the sample exhibited. The average grade is equal to D, and the median grade is equal to E. There are relatively few houses with grade A in the sample, around 1 percent and additional 6 percent with grade B. However, if we are investigating houses younger than 10 years, we can observe that more than 3 percent have grade A, and almost 19 percent have grade B. The median grade among homes built the last 10 years is C. Hence, this fact highlights the potential bias that might exist in the sample and force use of the propensity score method.

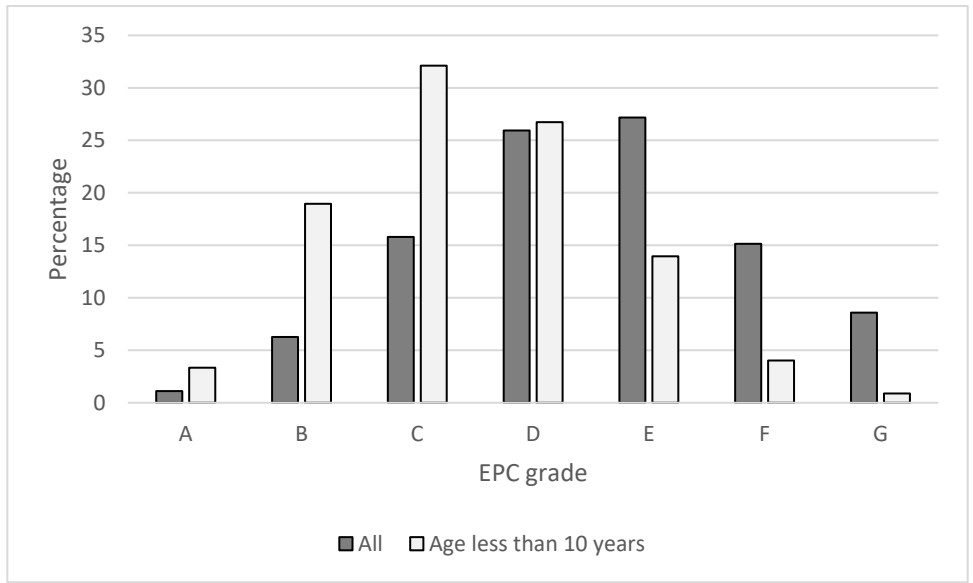

**Figure 2.** Frequency EPC grades.

Table 1 shows descriptive statistics concerning the dependent variable and the independent variables for the treatment group, the control group, and all.

**Table 1.** Descriptive statistics in the treatment and control group. Mean and standard deviation (within brackets).

|  | **Treatment** | **Control** | **Total** |
| --- | --- | --- | --- |
| Price (SEK) | 3,257,628 | 3,133,500 | 3,162,818 |
|  | (2,088,475) | (2,104,361) | (2,101,268) |
| Living area (Square meter) | 140 | 130 | 132 |
|  | (43) | (44) | (44) |
| Number of rooms | 5.50 | 5.15 | 5.23 |
|  | (1.42) | (1.49) | (1.48) |
| Age(years) | 45 | 52 | 51 |
|  | (28) | (27) | (28) |
| Plot size (square meter) | 2358 | 2199 | 2237 |
|  | (20,008) | (16,743) | (17,569) |
| Year-month | 2015-06 | 2016-06 | 2015-22 |
|  | (126) | (128) | (128) |
| Latitude | 58.10 | 59.10 | 58.64 |
|  | (2.42) | (2.42) | (2.23) |
| Longitude | 15.44 | 15.23 | 15.28 |
|  | (2.70) | (2.56) | (2.59) |
| Number of observations | 19,550 | 63,224 | 82,774 |

The price difference between the treatment group and the control group is small and not statistically significant. On the other hand, some of the housing attributes differ substantially between the treatment group and the control group. For example, the size of the houses in the treated sample is around 141 square meters compared to only 130 square meters in the control group. We can also observe that the average age is lower in the treatment group. That is, it is more likely that a new, larger house also has a higher EPC rating than an older and smaller home. This selection bias can potentially have some consequences when we estimate the EPC effect on housing prices.

*Step 3: Propensity score method*

In the first step of the propensity score matching, we estimated the propensity score with logistic regression where treatment is the dependent variable and a set of housing attributes such as living area, age, plot size, and number of rooms, as well as location attributes such as longitude, latitude, municipality, and county. Time is also included in the propensity score model. Figure 3 displays the propensity score of the logistic regression.

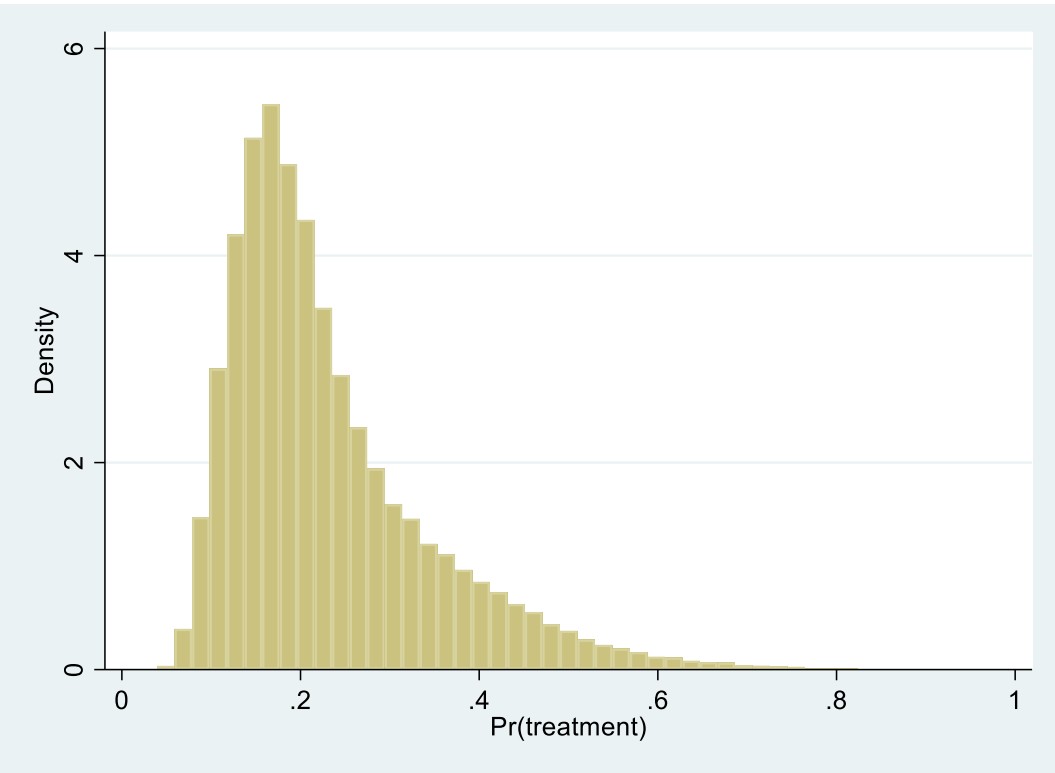

**Figure 3.** Propensity score, logistic regression.

The explanation power is on the lower side. However, the primary goal with the propensity score model is not to maximize the goodness to fit; the goal is to create as small differences between treated and untreated as possible. The authors of [27] proposed to start comparing the similarity between the groups in the matched sample by comparing means, medians, and standard deviations. In the following step, we therefore applied propensity score matching. As we said earlier, we performed two different types of matching, specifically, the nearest neighbor and radius. Table 2 exhibits the average values for some main attributes for the treated properties and control groups.

**Table 2.** Balancing property, matched samples using nearest neighbor and nearest neighbor within the radius. Average and standard deviation (within brackets).

|  | Nearest Neighbor | | Within Radius | |
|---|---|---|---|---|
|  | **Treated** | **Control** | **Treated** | **Control** |
| Age (years) | 45.94 | 49.17 | 51.85 | 51.20 |
|  | (27.71) | (27.60) | (24.42) | (26.92) |
| Living area (square meters) | 139.84 | 134.51 | 132.60 | 131.43 |
|  | (42.42) | (41.16) | (36.72) | (37.50) |
| Plot size (square meters) | 2271.13 | 2185.66 | 2216.00 | 2104.97 |
|  | (19,562) | (15,091) | (18,353) | (15,982) |
| Number of rooms | 5.50 | 5.32 | 5.32 | 5.23 |
|  | (1.42) | (1.44) | (1.30) | (1.34) |
| Year-month | 2015-12 | 2015-12 | 2015-12 | 2015-12 |
|  | (137.09) | (133.27) | (128.29) | (129.80) |
| Latitude | 59.11 | 58.77 | 58.54 | 58.54 |
|  | (2.42) | (2.24) | (2.06) | (2.07) |
| Longitude | 15.43 | 15.33 | 15.16 | 15.21 |
|  | (2.70) | (2.64) | (2.57) | (2.59) |
| Price (SEK) | 3,241,247 | 3,245,903 | 3,158,499 | 3,217,498 |
|  | (2,047,234) | (2,128,380) | (1,998,327) | (2,111,472) |
| Number of observations | 18,774 | 10,814 | 9312 | 7658 |

The first thing that we can observe is that there are small differences between the matching methods; they all produce almost the same matching transactions. Moreover, the number of treated transactions is nearly the same, and the control group is slightly smaller than the treated group. The total sample of matched transactions is considerably more limited than the total number of observations that we initially used. The difference in average values concerning the housing attributes is small. None of them are statistically significant at a 5% percent significance level. The difference in the matched sample with the unmatched sample is substantial. Compared with the statistics in Table 1, the matching procedure controls for all differences. Based on this, we are confident that we have balanced the data set effectively. Concerning the price, the difference in average price in the treated and the control group is not significant, and more surprisingly, the difference has the wrong sign.

*Step 4: Hedonic price equation*

The next step is to run the hedonic regressions. The default model is a model where the outlier and selection bias are uncontrolled. The second and third models control for outlier and selection bias using influential weights and the propensity score directly into the hedonic price equation. We first included the propensity score as a covariate (model name Multivariate 1) and using the inverse propensity score as sample weights (Multivariate 2). Next, we controlled for selection bias by only using the matched sample, namely, by the nearest neighbor (Matched 1) and within a radius (Matched 2). Finally, we controlled for selection bias using the estimated strata as fixed effects (Stratified). Table 3 exhibits the results from all the above models. All models were estimated as log-linear models where the natural logarithmic of the price is the dependent variable, and the independent variables' living area and age are also natural logarithmic, while the rest of the independent variables are untransformed.

**Table 3.** The preliminary hedonic price equation and with unmatched and matched samples.

| | Default | Multivariate (1) | Multivariate (2) | Matched (1) | Matched (2) | Stratified |
|---|---|---|---|---|---|---|
| EPC | 0.0514 | 0.0368 | 0.0410 | 0.0331 | 0.0356 | 0.0371 |
| | (17.66) | (16.72) | (17.19) | (10.29) | (8.76) | (16.91) |
| Ln(Living area) | 0.5925 | 0.5502 | 0.5603 | 0.5763 | 0.5445 | 0.5282 |
| | (108.75) | (116.06) | (76.83) | (76.81) | (54.50) | (96.42) |
| Number of rooms | 0.0359 | 0.0243 | 0.0431 | 0.0398 | 0.0444 | 0.0303 |
| | (32.41) | (22.62) | (23.50) | (26.99) | (21.69) | (22.72) |
| Plot size | 0.0001 | 0.0001 | 0.0001 | 0.0001 | 0.0001 | 0.0001 |
| | (24.53) | (26.46) | (7.48) | (16.40) | (10.45) | (26.80) |
| Ln(Age) | −0.0998 | −0.0404 | −0.0801 | −0.0879 | −0.0817 | −0.0472 |
| | (−56.05) | (−12.43) | (−46.58) | (−45.20) | (−21.52) | (−12.35) |
| Propensity score | - | 0.5042 | - | - | - | - |
| | | 16.27 | | | | |
| Constant | 55.4015 | 58.0831 | 71.8171 | 73.3430 | 76.7408 | 76.7566 |
| | (47.32) | (46.70) | (45.02) | (43.37) | (31.95) | (73.77) |
| Fixed strata effect | No | No | No | No | No | Yes |
| Sample weights | No | No | Yes | No | No | No |
| Fixed county and municipality effects | Yes | Yes | Yes | Yes | Yes | Yes |
| Fixed time effects | Yes | Yes | Yes | Yes | Yes | Yes |
| $R^2$ adjusted | 0.7507 | 0.8636 | 0.8653 | 0.8553 | 0.8489 | 0.8581 |
| Shapiro-Francia (*p*-value) | 0.0000 | 0.0000 | 0.0000 | 0.0000 | 0.0001 | - |
| Breusch-Pagan (*p*-value) | 0.0000 | 0.0000 | - | 0.0000 | 0.0000 | - |
| VIF (Treatment) | 1.06 | 1.13 | 1.12 | 1.08 | 1.05 | - |
| No. of observations | 99,877 | 80,260 | 80,260 | 29,588 | 16,970 | 80,260 |

Note. In the Multivariate (s) model, the propensity score is used as an independent variable. In the Multivariate (2) model, the inverse of the propensity score is used as sample weights. The stratified modeling approach uses estimated strata from the propensity score rating as fixed effects.

Overall goodness-of-fit (adjusted $R^2$) is around 75–87 percent, which can be considered to be excellent and comparable to other hedonic price studies. The difference between the default model and the models controlling for outlier and selection bias is substantial. Estimated coefficients concerning living area, number of rooms, and age are of reasonable magnitude, and they are all statistically significant. For example, one extra room is expected to increase the price by around 3 percent, and a year-older, single-family house will decrease the price by 0.16 percent. The estimates are also robust across specifications of the hedonic model. The EPC variable is estimated to be at its lowest at 3.36 percent (coefficient 0.033) in the Matched model 2 compared to 5.27 percent (coefficient 0.0514) in the Default model. That is equivalent to almost SEK 100,000 (Euro 9278) and approximately SEK 200,000 (Euro 18,556), respectively. Hence, failure to consider outliers and potential selection problems can have significant consequences and policy implications.

The low VIF (variance inflation factor) value concerning our EPC variable indicates that the multicollinearity problem is modest. Moreover, we can reject the null hypotheses of heteroscedasticity (Breusch–Pagan test) and non-normality (Shapiro–Francia test), respectively.

*Step 5: Quantile regression*

Using the propensity score matching and the quantile regression, we can analyze if the treatment effect varies in the price distribution. Table 4 shows the estimated treatment effect, presented together with t-values in each percentile of the price distribution.

It is interesting to observe that the impact of high energy performance on the house prices does not vary as much as expected. It seems that our results do not support the findings of, for example, [14]. Our results indicate the impact of EPC is present in all housing price segments and that the percentage impact is almost the same. The effect only varies between 2.62 and 2.94 percent.

*Step 6: Robustness of the model*

Spatial dependency may be present even if we are controlling for space by including location covariates, such as fixed effects for the municipality and the county, as well as longitude and latitude. Our results are robust, tested by controlling for spatial dependence. Two different spatial weight matrices were used, namely, inverse distance and inverse distance within 4 kilometers. We estimated a spatial error model (SEM), a spatial autoregressive model (SAR), and a spatial Durbin model (SDM).

We estimated the models with maximum likelihood estimations. The total effects of the various models are presented in Table 5.

**Table 4.** Treatment impact (quantile regression). Matched sample.

| Percentile | Coefficient | t-Value | Impact (%) |
|---|---|---|---|
| 0.9 | 0.0269 | 7.03 | 2.73 |
| 0.8 | 0.0277 | 7.42 | 2.81 |
| 0.7 | 0.0284 | 7.81 | 2.88 |
| 0.6 | 0.0290 | 8.15 | 2.94 |
| 0.5 | 0.0270 | 7.31 | 2.74 |
| 0.4 | 0.0266 | 7.10 | 2.70 |
| 0.3 | 0.0277 | 7.20 | 2.81 |
| 0.2 | 0.0285 | 7.25 | 2.89 |
| 0.1 | 0.0259 | 5.37 | 2.62 |

**Table 5.** Spatial error model, spatial autoregressive model, and spatial Durbin model (direct effects). Matched sample. Maximum likelihood estimates.

| | SEM | | SAR | | SDM | |
|---|---|---|---|---|---|---|
| | **W1** | **W2** | **W1** | **W2** | **W1** | **W2** |
| EPC | 0.0342 | 0.0342 | 0.0337 | 0.0340 | 0.0340 | 0.0343 |
| | (5.43) | (5.43) | (5.35) | (5.39) | (5.39) | (5.43) |
| Ln(Age) | −0.0844 | −0.0844 | −0.0837 | −0.0837 | −0.0847 | −0.0849 |
| | (−14.33) | (−14.34) | (−14.23) | (−14.22) | (−14.32) | (−14.35) |
| Ln(Living area) | 0.5460 | 0.5460 | 0.5474 | 0.5473 | 0.5470 | 0.5472 |
| | (35.08) | (35.08) | (35.18) | (35.18) | (35.15) | (35.16) |
| No. of rooms | 0.0453 | 0.0453 | 0.0453 | 0.0453 | 0.0451 | 0.0452 |
| | (14.33) | (14.33) | (14.32) | (14.31) | (14.26) | (14.27) |
| Plot size | 0.0001 | 0.0001 | 0.0001 | 0.0001 | 0.0001 | 0.0001 |
| | (9.41) | (9.41) | (9.37) | (9.36) | (9.22) | (9.07) |
| Constant | 81.3160 | 81.3747 | 81.45453 | 81.5450 | 81.3110 | 81.4391 |
| | (20.81) | (24.69 | (23.06) | (27.49) | (19.70) | (23.43) |
| Rho | | | −0.0009 | −0.0030 | | |
| | | | (−0.24) | (−0.79) | | |
| Lamda | 0.7491 | 0.7087 | | | | |
| | (6.26) | (6.57) | | | | |
| Fixed time effect | Yes | Yes | Yes | Yes | Yes | Yes |
| Fixed county effect | Yes | Yes | Yes | Yes | Yes | Yes |
| Fixed municipality effect | Yes | Yes | Yes | Yes | Yes | Yes |
| Wald test (*p*-value) | 0.0000 | 0.0000 | 0.8134 | 0.4271 | 0.0000 | 0.0000 |
| Pseudo R2 | 0.8290 | 0.8290 | 0.8290 | 0.8428 | 0.8294 | 0.8293 |

The first thing we notice is that our estimate concerning the EPC is remarkably robust across spatial econometric specifications. The estimates only vary from 3.43 percent to 3.49 percent (not significantly different from each other). Moreover, spatial dependency is present, but it will not cause any bias or estimates. Third, the spatial component does not add to the goodness-of-fit. Hence, we concluded that the spatial dependency in the hedonic model does not have any severe impact on our estimates.

*Step 7: Parameter heterogeneity*

Sweden is a long country from the 55th-degree to 68th-degree latitude (that is, closer to the North Pole than to the equator). Sweden has two different climatic zones, namely, the subarctic climate zone (from 60 degrees latitudes and up) and the hemiboreal climate zone. Figure 4 shows a map of Sweden and the differences in annual average temperatures. In the southern part of Sweden, the yearly temperature is around 7–10 degrees Celsius, and up in the north, it is around 0 degrees. The climate is warmer along the east coast than inland. Lower temperatures up in the subarctic zone could potentially

mean that EPC impacts house prices more as energy costs are more likely to be higher. As shown in [15], energy costs have an impact on people's willingness to pay for high EPC ratings in Finland. To test this hypothesis, we created an interaction variable between treatment effect and a binary variable that indicates latitudes higher than 60 degrees—that is, close to the subarctic climate zone.

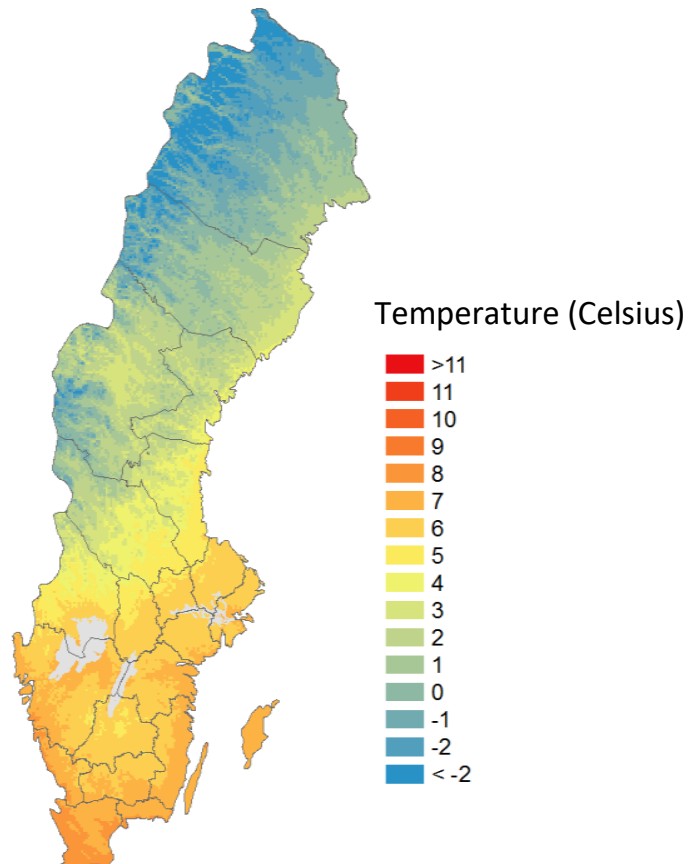

**Figure 4.** A map of Sweden with an average annual temperature (Celsius).

If the estimated parameter concerning the interaction variable is significant, we can accept the hypothesis that there is more willingness to pay for the high rating in the subarctic climate zone. Table 6 displays the results. The results also indicate that the impact of EPC on house prices in the northern part of Sweden (5 percent) is higher than in the southern region (3 percent). Regardless of the matching method, the estimates are significantly different from zero, indicating that we should unsurprisingly reject the hypothesis that energy expenses do not have an impact on capitalization. Hence, climate influences energy cost and, therefore, potentially could have an impact on the willingness to pay for high EPC ratings. Something we have not tested here is whether the capitalization has a seasonal effect. An expected impact of season could be that a high EPC rating in northern Sweden is capitalized more during the winter months than in the summer. It is an issue that is well suited to future research.

**Table 6.** Parameter heterogeneity. Matched sample and stratified sample.

|  | Interaction | South | North | Stratified |
|---|---|---|---|---|
| EPC | 0.0316 | 0.0261 | 0.0693 | 0.0347 |
|  | (9.69) | (7.91) | (7.23) | (15.37) |
| EPC-north | 0.0208 | - | - | 0.0315 |
|  | (2.78) |  |  | (4.71) |
| Ln(Living area) | 0.5758 | 0.5664 | 0.6117 | 0.5277 |
|  | (76.72) | (72.44) | (28.91) | (96.32) |
| Number of rooms | 0.0398 | 0.0401 | 0.0366 | 0.03028 |
|  | (27.01) | (26.29) | (8.63) | (22.73) |
| Plot size | 0.0001 | 0.0001 | 0.0001 | 0.0001 |
|  | (10.45) | (18.70) | (6.81) | (26.76) |
| Ln(Age) | −0.0879 | −0.0856 | −0.1105 | −0.0474 |
|  | (−45.23) | (−43.67) | (−16.62) | (−12.40) |
| Constant | 75.4048 | 76.5737 | 65.6259 | 76.7954 |
|  | (43.41) | (38.37) | (16.74) | (73.81) |
| Fixed strata effect | No | No | No | Yes |
| Fixed county and municipality effects | Yes | Yes | Yes | Yes |
| Fixed time effects | Yes | Yes | Yes | Yes |
| $R^2$ adjusted | 0.8553 | 0.8502 | 0.7878 | 0.8582 |
| No. of observations | 29,588 | 23,995 | 5,633 | 80,253 |

## 4. Discussion and Conclusions

The impact of energy performance certificates on housing prices has been investigated extensively in recent years. However, the results from these investigations have been mixed. We attempted to add to the literature by more specifically controlling for different types of biases. We used a slightly different approach to estimate the causal relationship between house prices and energy performance certificates. We used the traditional hedonic modeling approach. However, we also utilized the propensity score method approach to be able to compare treated houses with a control group. We also investigated the importance of outliers, spatial dependency, and parameter heterogeneity of our estimates. Moreover, we used a quantile regression technique to test the hypothesis that the capitalization effect varies across the price distribution.

Our main results indicate there is an upward bias in the estimated parameter concerning EPC if we are not controlling for outliers and selection bias. Regardless of the propensity score method approach, the results are lower than the default model, not controlling for influential observations and selection bias. The impact is substantial: 3 percent compared to 6 percent in the default model, not controlling for various kinds of biases. Our estimated capitalization effect is around 3 percent, which is lower than, for example, that in [8–11]. On the other hand, compared to [10], we estimated a statistically significant positive impact on the Swedish housing market.

Moreover, our results do not support that the impact of energy performance certificates varies in the price distribution. Hence, the certificates are not differently capitalized in the high-end housing price segment. Finally, our results support the hypothesis that the energy performance certificate is more greatly capitalized in the northern and colder parts of Sweden supporting expectations and the results, for example, of [15].

What policy implications do the results have? The goal of the introduction of EPC was to convince homeowners to pay attention to the energy consumption of their house, and as an effect, to take measures to reduce their dwelling's climate impact. The aim was also to provide prospective buyers with information to make rational choices in this regard. Since, in this study, we can observe the capitalization of EPC in property values, it can be inferred that the reform has had the desired effect. Hence, this should ultimately lead to a reduction in energy consumption in the housing sector.

Is the capitalization complete? The capitalization effect is at a level comparable to the referenced studies, but is it sufficient for the ownership of dwellings to carry out financially justified investments? There are indications that the estimated effect is on the low side, given today's energy prices. Thus, to clarify and increase the visibility of the EPC rating and to offer a better explanation of what it means in the description of a house for sale would probably result in providing potential buyers with even better information than now. An interesting experiment for future research would be to get a real estate agent to do just that for some of the agency's properties that are for sale. If this were done randomly, it would allow us to estimate the causal impact of the improved information on EPC on house prices.

**Funding:** This research was funded by the research project Bostad 2.0 (Housing 2.0) at the Royal Institute of Technology (KTH).

**Acknowledgments:** Mäklarstatistik AB generously provided the utilized data.

**Conflicts of Interest:** The author declares no conflict of interest.

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
