# Peer review of "Energy Performance Certificates and Its Capitalization in Housing Values in Sweden"

_sustainability, doi:10.3390/su11216101_

Round 1
Reviewer 1 Report
The subject is interested, and there is a very good efforts I see it through my reading in this important paper.
I recommend to published it with minor revision:
Some changes in the structure of paper: in line 116 section "Theoretical framework and used methodology", my suggestion to separate "theoretical frame work" in one section and the "Data and Methodology" in other section. and give more details about the data. It's important to add the size of data in the abstract of the paper. I recommend to add the program that you used in the paper. In the model: you used logarithm price for dependent variable, I think it's good to try to use logarithm Living area and the logarithm age in independent variables. You used data along the different day and months during three years, so the temperature effect I think is important, can you describe how you fixed this in your model.
Author Response
Comments to the reviewers.
First of all, thank you for good comments. I have tried to handle them all.
Review Report Form
Open Review
(x) I would not like to sign my review report
( ) I would like to sign my review report
English language and style
( ) Extensive editing of English language and style required
( ) Moderate English changes required
(x) English language and style are fine/minor spell check required
( ) I don't feel qualified to judge about the English language and style
|
Yes |
Can be improved |
Must be improved |
Not applicable |
|
|
Does the introduction provide sufficient background and include all relevant references? |
(x) |
( ) |
( ) |
( ) |
|
Is the research design appropriate? |
( ) |
(x) |
( ) |
( ) |
|
Are the methods adequately described? |
( ) |
(x) |
( ) |
( ) |
|
Are the results clearly presented? |
(x) |
( ) |
( ) |
( ) |
|
Are the conclusions supported by the results? |
(x) |
( ) |
( ) |
( ) |
Comments and Suggestions for Authors
The subject is interested, and there is a very good efforts I see it through my reading in this important paper.
I recommend to published it with minor revision:
Some changes in the structure of paper: in line 116 section "Theoretical framework and used methodology", my suggestion to separate "theoretical frame work" in one section and the "Data and Methodology" in other section. and give more details about the data. Yes, changed the name to Methodology as this describes the section best.
It's important to add the size of data in the abstract of the paper. Yes, it is now included.
I recommend to add the program that you used in the paper. Yes, I have included that Stata is used.
In the model: you used logarithm price for dependent variable, I think it's good to try to use logarithm Living area and the logarithm age in independent variables. Yes, a very good comment that had big effect!! I did check and found out that ln of living area and age is preferred. I have therefore changed all results in the tables. In the overall model, nothing changed. But it did get big effect on the north-south model. Now there is a clear north effect. The willingness to pay (as a percentage of house prices) for a high grade in the north is double compared to that in the south.
You used data along the different day and months during three years, so the temperature effect I think is important, can you describe how you fixed this in your model. Not handled at all, but this seasonal effect could be handled by including dummy variables.
Submission Date
14 September 2019
Date of this review
05 Oct 2019 12:23:24
Reviewer 2 Report
The author presents an interesting analysis and calculations which are reasonably clearly discussed. The author still needs to clearly state the importance of the obtained results, for example:
Who is the potential user of this information? Why is this information relevant to a global reader (not from Sweden)? How do the findings of this work can impact on the EPC's, energy regulations in Sweden and worldwide? Finally, from the obtained results, it was found that the results do not support that the impact of EPC's varies in the price distribution. So, how does the author envisions that the EPC's could be successfully capitalized within the Swedish housing market? From the findings of this work, what are the future analysis in this direction?
General comments:
English spelling is strongly encouraged Revise that the font of tables is coherent with the manuscript font Please provide the references to the figures that help sustain the introduction of the work and revise that all of the references included in the text are listed on the Reference section The manuscript sections need some rearrangement in order to comply with the journal style, for example:
-Section 2. Theoretical framework and used methodology. This section should be renamed to Methodology or Methods
-Section 3. Empirical analysis. This section is related to the results and discussion of the results. It is recommended that this section is divided into a "Results" and a "Discussion" section. A discussion section is needed.
Review the reference style of the Journal.
Also, a thorough revision of the Instructions for Authors of the Journal is strongly encouraged.
Specific comments:
Line 24: "The importance of housing on climate change…" Please state if the author means "housing in general" or is directly referring to the "housing energy consumption"
Line 25: The sentence referring to the building sector accounts for as much as 40% of the total energy consumption, needs a reference.
Line 28: Add a definition of "Eurostat" and reference to the provided information.
Line 30: The sentence containing information about 65% of final energy consumption... needs to be referenced. Also, about this statement, please mention the weather/location or country… from which this statistic was taken? Due to the percentage of heating/cooling energy requirements is intensely related to the weather where the house is located.
Line 34: the reference from Eckartz, 2016 is not listed in the reference section.
Line 36: the sentence "… and 5 percent of that is due to energy cost" needs a reference.
Line 70: The reference Fuesta, 2016 has a typo. Please correct the spelling.
Line 116: On this section, the paragraphs related to the theoretical framework should be moved to the introduction.
Line 117: The reference from Rosen (1974) and Rousseeuw and Leroy (1987) are not listed in the references section.
Line 237: Please state for a wider audience what kind of institution or business is Mäklarstatistic AB.
Figure 2. Add the legend of y-axis on the figure.
Table 1. The author should add the currency of the housing prices.
Tables 1-2-3. The author should include all of the units of the given parameters (living area, age, plot size,…)
Line 340: Describe "VIF",. Is VIF related to the "variance inflation factor" concept?
Figure 5: Change "temperatur" to English, add a North figure/sign for reference and a scale to the map.
Author Response
First of all, thank you for good comments. I have tried to handle them all.
Review Report Form
Open Review
(x) I would not like to sign my review report
( ) I would like to sign my review report
English language and style
(x) Extensive editing of English language and style required
( ) Moderate English changes required
( ) English language and style are fine/minor spell check required
( ) I don't feel qualified to judge about the English language and style
|
Yes |
Can be improved |
Must be improved |
Not applicable |
|
|
Does the introduction provide sufficient background and include all relevant references? |
( ) |
(x) |
( ) |
( ) |
|
Is the research design appropriate? |
( ) |
(x) |
( ) |
( ) |
|
Are the methods adequately described? |
( ) |
( ) |
(x) |
( ) |
|
Are the results clearly presented? |
( ) |
(x) |
( ) |
( ) |
|
Are the conclusions supported by the results? |
( ) |
( ) |
(x) |
( ) |
Comments and Suggestions for Authors
The author presents an interesting analysis and calculations which are reasonably clearly discussed. The author still needs to clearly state the importance of the obtained results, for example:
Who is the potential user of this information? Why is this information relevant to a global reader (not from Sweden)? How do the findings of this work can impact on the EPC's, energy regulations in Sweden and worldwide? New section is included in the introduction.
Finally, from the obtained results, it was found that the results do not support that the impact of EPC's varies in the price distribution. So, how does the author envisions that the EPC's could be successfully capitalized within the Swedish housing market? A new section is included in the final discussion section.
From the findings of this work, what are the future analysis in this direction? A new section is included in the final discussion section.
General comments:
English spelling is strongly encouraged Proofreading is done.
Revise that the font of tables is coherent with the manuscript font Please provide the references to the figures that help sustain the introduction of the work and revise that all of the references included in the text are listed on the Reference section The manuscript sections need some rearrangement in order to comply with the journal style, for example: Ok
-Section 2. Theoretical framework and used methodology. This section should be renamed to Methodology or Methods Yes, has been done
-Section 3. Empirical analysis. This section is related to the results and discussion of the results. It is recommended that this section is divided into a "Results" and a "Discussion" section. A discussion section is needed.I have added a discussion into the conclusion section. New title: Discussion and conclusion
Review the reference style of the Journal.Yes, has been done.
Specific comments:
Line 24: "The importance of housing on climate change…" Please state if the author means "housing in general" or is directly referring to the "housing energy consumption" a general statement. Included both of our statements.
Line 25: The sentence referring to the building sector accounts for as much as 40% of the total energy consumption, needs a reference.Yes, included
Line 28: Add a definition of "Eurostat" and reference to the provided information. Yes, included
Line 30: The sentence containing information about 65% of final energy consumption... needs to be referenced. Also, about this statement, please mention the weather/location or country… from which this statistic was taken? Due to the percentage of heating/cooling energy requirements is intensely related to the weather where the house is located.Yes, included
Line 34: the reference from Eckartz, 2016 is not listed in the reference section.deleted
Line 36: the sentence "… and 5 percent of that is due to energy cost" needs a reference.Yes included
Line 70: The reference Fuesta, 2016 has a typo. Please correct the spelling. Changed
Line 116: On this section, the paragraphs related to the theoretical framework should be moved to the introduction.Do not understand
Line 117: The reference from Rosen (1974) and Rousseeuw and Leroy (1987) are not listed in the references section. Yes, included
Line 237: Please state for a wider audience what kind of institution or business is Mäklarstatistic AB.Yes, has been done
Figure 2. Add the legend of y-axis on the figure.Yes, done in one figure
Table 1. The author should add the currency of the housing prices.Yes, has been done
Tables 1-2-3. The author should include all of the units of the given parameters (living area, age, plot size,…) Yes, has been done
Line 340: Describe "VIF",. Is VIF related to the "variance inflation factor" concept? Yes
Figure 5: Change "temperatur" to English, add a North figure/sign for reference and a scale to the map.Yes, has been done
Round 2
Reviewer 2 Report
The comments made to the contributions section, conclusions and the overall scope of the work were clearly stated and are a great help to understand the value of the work.
The author made most of the recommended changes to the manuscript. Some of them were marked in yellow at the document. Nevertheless, the changes were not so easy to follow.
At the response document the author responded that he attended the recommendations, but, for example, one of the changes that it was marked as attended was not made to the document. The Tables with parameters/results do not have the corresponding units (m2, $...).
It is suggested that the author sends a more detailed response document where it is pointed out, either the lines where the change is made or the change is pasted next to the recommendation.
Author Response
So sorry about that. The new tables 1 and 2 were by mistake shifted back to the old ones after proof reading. In table 1 and 2 I have given the measurement unit for Price, Living area, Plot area and Age.
Sorry if the changes was difficult to follow. First I did use "track changes" but it became difficult to see the changes especially after the proof reading and change of reference system. I marked it in yellow the major changes. Then I have made small changes in different parts of the text. Yes, maybe I should have been more careful in the reply to the reviewer.
Thank you for your careful reading.
Best regards